# Scaling-up the Resources for a Freely Available
# Swedish VADER (*svVADER*)

**Dimitrios Kokkinakis**
University of Gothenburg and Centre for
Ageing and Health (AgeCap)
dimitrios.kokkinakis@gu.se

**Ricardo Muñoz Sánchez**
University of Gothenburg
ricardo.munoz.sanchez@gu.se

**Mia-Marie Hammarlin**
Lund University and Birgit Rausing
Centre for Medical Humanities (BRCMH)
mia-marie.hammarlin@kom.lu.se

## Abstract

With widespread commercial applications in various domains, sentiment analysis has become a success story for Natural Language Processing (NLP). Still, although sentiment analysis has rapidly progressed during the last years, mainly due to the application of modern AI technologies, many approaches apply knowledge-based strategies, such as lexicon-based, to the task. This is particularly true for analyzing short social media content, e.g., tweets. Moreover, lexicon-based sentiment analysis approaches are usually preferred over learning-based methods when training data is unavailable or insufficient. Therefore, our main goal is to scale-up and apply a lexicon-based approach which can be used as a baseline to Swedish sentiment analysis. All scaled-up resources are made available, while the performance of this enhanced tool is evaluated on two short datasets, achieving adequate results.

## 1 Introduction

Sentiment analysis is the computational study of people's opinions, sentiments, emotions, and attitudes towards entities such as products and services, and their attributes. Sentiment analysis allows tracking of the public's mood about a particular entity to create actionable knowledge (Ligthart et al., 2021) and has found numerous applications, ranging from digital humanities (Kim & Klinger, 2022) to gaining insight into customers' feedback about commercial products and services (Rashid & Huang, 2021). Sentiment analysis can occur at the document, sentence, or word level,

while the sentiment types usually assigned are *Very positive*, *Positive*, *Neutral*, *Negative* or *Very negative*. E.g., the sentiment for the sentence *Att känna stödet från publiken och folket är väldigt smickrande* 'To feel the support of the audience and the people is very flattering' will be usually assigned a positive sentiment while the sentence *Föräldrar i chock efter bluffen i basketlaget* 'Parents in shock after the hoax in the basketball team' will be assigned a negative one.

In this paper we discuss an enhancement of a popular off-the-shelf (unsupervised) dictionary-based approach to Sentiment analysis using VADER (*Valence Aware Dictionary and sEntiment Reasoner*). VADER is a lexicon and rule-based sentiment analysis tool that is specifically attuned to sentiments expressed in social media (Hutto & Gilbert, 2014). VADER is fully open-sourced, available e.g., from the NLTK package (Bird et al., 2009), which can be applied directly to unlabeled text data. Furthermore, VADER can efficiently handle large vocabularies, including the use of degree modifiers and emoticons. These qualities make VADER a good fit for use on social media input for rapid sentiment text analysis. As such, the need for previous training as in machine or deep learning models, is eliminated.

Our main aim of this work is to make VADER a useful baseline for Swedish sentiment analysis, by rapidly scaling-up and improving the coverage of the already translated to Swedish resources (lexicons and processing tools). We further evaluate the coverage by applying and comparing the original VADER translation with the enhanced version on two small datasets, one with Swedish tweets and one sample from the *ABSAbank-Imm*[1], an annotated Swedish corpus for aspect-based se-

---

[1] https://spraakbanken.gu.se/en/resources/absabank-imm.

ntiment analysis (Rouces et al., 2020).

## 2 VADER

The *Valence Aware Dictionary for sEntiment Reasoning* (VADER[2]) is a parsimonious rule-based model for sentiment analysis of specifically social media text (Hutto & Gilbert, 2014). Since its release VADER has been extensively used in various applications and domains; from the analysis stock news headlines (Nemes & Kiss (2021); to the assessment of sentiments expressed in customers' e-mails (Borg and Boldt, 2020); and further to the analysis of tweets on COVID-19 vaccine hesitancy (Verma et al., 2022).

### 2.1 VADER translations

VADER lexical components have been translated into several languages, such as German, French, and Italian[3]. The Swedish translation of the VADER sentiment lexicon, along with the VADER application's negators and booster words, were translated from English to Swedish, using the Google Cloud Translation API by Gustafsson (2019). However, one third of the original English sentiment lexicon remained untranslated during this process, which in a sense decrease the quality of the analysis. According to Gustafsson (2019) the original English VADER lexicon contained 7517 words, slang words, abbreviations, and emoticons. Out of these, 2435 could not be translated to Swedish because no translation could be found; for instance, many words in the English lexicon had inflections that did not exist in the Swedish counterpart; polysemy created problems, as well as English idiomaticity, e.g., slang words remained untranslated. This version of the Swedish VADER sentiment resources can be found in Github[4].

### 2.2 Enhancements of the translated Swedish VADER: *single words, lexicalized idioms, and other multiword expressions*

The original Swedish translation of VADER was the starting point for developing and enhanced version of VADER (svVADER[5]). In general, VADER is based on a few key points when determining the sentiment of a text:

- *degree modifiers or booster words*, that is dampeners and intensifiers, i.e., words or characters that affects the magnitude of the polarity by either increasing or decreasing the intensity of the sentiment;
- *negations,* words which reverse the semantic orientation in a text and thus also its polarity score;
- *capitalization*, which increases the intensity of polarity, and the sentiment becomes intensified, and,
- certain types of *punctuation*, specifically exclamation marks which increase the intensity of polarity without affecting the semantic feeling.

We started refining and adapting the VADER script, in which booster words and negation items are hard coded. We both added new booster lexical items (e.g., *knappast*; *minimalt*; *svagt*; and *måttligt*) and deleted several dubious words (e.g., *effing*; *flippin*; *frackin*; *fuggin* and *hella*); similarly, some missing Swedish negation words (e.g., *icke*; *inget*; *inga* and *ej*) were also added to this script.

The characterization of the multiword expressions (MWE) and their idiomaticity play an important role in lexically based sentiment analysis. For instance, Moreno-Ortiz et al. (2013) discuss that MWEs, being units of meaning, their relative weight to the calculated overall sentiment rating of texts needs to be accounted for as such, rather than the number of component lexical units. Therefore, we added a list of 100 sentiment laden idioms, that is multiword expressions the meaning of which cannot be deduced from the literal meaning of constituent words (e.g., the Swedish idioms *blåst på konfekten* 'to be cheated on' and the Swedish idiom *tomtar på loftet* which is used to refer to someone who is stupid or crazy). The lexicalized idioms originate from the available list of the NEO lexicon DB[6] that contains a large number (over 4,000) of lexicalized idioms; the selection was made by matching all items on Tweeter and Flashback corpora, extracting the matches, and browsing manually the matched idioms annotating relevant items as positive or negative. Moreover, we manually annotated and

---

[2] github.com/cjhutto/vaderSentiment.
[3] See here German (Tyman et al., 2019) (github.com/KarstenAMF/GerVADER); French (github.com/vr0nsky/vadersentiment_fr) and here details for Italian (Martinis et al., 2022).
[4] github.com/AlexGustafsson/vaderSentiment-swedish.
[5] github.com/XdimitrisX/svVADER.
[6] spraakbanken.gu.se/en/resources/neo-idiom.

added over 200 phrasal verbs, (e.g., *spränga ihjäl sig* 'to blow yourself up'; *skälla ut* 'to scold'; *rusta ner* 'to gear down' and *rusta upp* 'to gear up'). Statistically significant collocations were also added, these were extracted from the analysis of the two larger collections where the two datasets originate from (cf. Section 3)*. Also, common medical terminology[7] (i.e., roughly 500 symptoms and frequent disease names) where added with negative polarity to svVADER's main lexicon. Finally, we created an emoj[8] list (3,500) with Swedish expansion (meaning) downloaded and refined from various Internet sites, i.e., 😂 *ansikte med glädjetårar* 'face with tears of joy'.

Table 1 shows the current lexical content of the original and enhanced versions of svVADER.

| Name | Size | License |
|---|---|---|
| Original translation | 5,501 | MIT License |
| Enhanced *single words* | 58,070 | CC BY 4.0* |
| Enhanced *MWE* | 2,300 | |

Table 1: The size of the Swedish lexicons (single words: includes inflected forms; MWE: Multi-Word Expressions; '*': license of the enhanced lexicons).

# 3 Application scenario: Swedish tweets about mRNA vaccines and Flashback posts on immigration

As an application scenario for the evaluation of svVADER we selected two small datasets. The first one consists of Swedish tweets posted in 2022 that discuss vaccine skepticism, and particularly, anxiety about possible side effects and concerns related to novel vaccine technologies, such as the messenger RNA (mRNA) which has be used as a reason for not receiving (the COVID-19) vaccine (Leong et al., 2022). The extracted Swedish tweets were collected with the keywords m-?RNA.* ('?' the preceding character is optional.; '.*': ≥ 0 characters) or the hashtag #mRNA and lang:sv (Swedish content). From the extracted tweets (ca 1,800), a random selection of 200 tweets was selected for the svVADER evaluation. The second dataset originates from the ABSAbank-Imm (where ABSA stands for "Aspect-Based Sentiment Analysis" and Imm for "Immigration", a subset of the Swedish ABSAbank) annotated dataset

(Rouces et al., 2020) where we randomly extracted 315 posts. ABSA models predict the sentiment of specific aspects present in the text, that is sentiment expressions that contain no polarity markers but still convey clear human-aware sentiment polarity in context (Russo et al., 2015). In ABSAbank-Imm, texts and paragraphs are manually labelled according to the sentiment (on 1-5 scale) that the author expresses towards immigration in Sweden (a task also known as stance analysis). The 315 posts come from the Flashback Forum[9], a popular Swedish discussion platform. For simplicity, the extracted posts consisted of posts with 1-2 sentences; posts that consisted of 3 or more sentences were excluded. Moreover, the selected posts were labelled as positive if their manually assigned score in ABSA was 5.0 (very positive) or 4.0 (positive) and negative if their manually assigned score was 1.0 (very negative) or 2.0 (negative). Posts that lied in the middle scale with ratio 3.0 were labelled as neutral. Thus, for practical reasons, we collapsed the scores 5.0 and 4.0 to positive sentiment and 1.0 and 2.0 to negative.

## 3.1 Experimental results and evaluation

The ABSAbank-Imm dataset was already manually labelled, while the Tweeter dataset was manually labelled by one of the authors and a Master student, the inter-annotator agreement[10] was high (Fleiss' $\kappa \approx 0.839$).

VADER's sentiment score is returned in both as a compound score or as *pos*itive, *neg*ative, and *neu*tral. The compound score is computed by summing the valence scores of each word in the text, adjusted according to the rules, and then normalized to be between –1 (very negative) and +1 (very positive). Specifically, VADER's compound sentiment score determines the underlying sentiment of a text (i.e., tweet or post) according to the following schema:

- positive, compound score $\geq 0.05$
- negative, compound score $\leq -0.05$
- a neutral, the compound score is between $> -0.05$ and $< 0.05$

We use the *original* Swedish VADER translation to automatically classify each tweet and each Flashback post according to its semantic

---

[7]Motivated by the fact that there is a growing interest to analyzed social media with health-related content.
[8]https://emojipedia.org/sv/.

[9]https://www.flashback.org/.
[10]For the interrater reliability and agreement, we applied the R package *irr* 0.84.1.

orientation and we then proceed to classify the same data with the enhanced resources. Table 2 summarizes the results of the evaluation, which clearly shows, as expected, that the enhanced approach improved the compound score results based on the original VADER translation.

| Modell | CS: Tweets | F1 ABSA |
|--------|-----------|---------|
| VADER | 36,7% | 37,2% |
| svVADER$_{single\ words}$ | 50,8% | 48,2% |
| svVADER$_{all}$ | 51% | 48,1% |

Table 2: Evaluation results for the original Swedish translation of VADER and the enhanced flavors of svVADER. (CS: Compound Score; svVADER$_{single\ words}$: original plus *new* non-MWE words).

For the evaluation of (sv)VADER's performance, we apply a slightly adopted version of the SemEval-2017 Task 4 (Rosenthal et al., 2017), evaluation script[11]. As with other approaches to sentiment analysis there are several pros and cons to the task. The approach is relatively easy to implement and understand, and, given the magnitude of customer experience for products and services available online it becomes doable to capture relevant datasets. However, since the model is primarily designed for use with social media content in mind, the analysis may easily overlook important words or usage. Social media input is usually loaded with typos, misspellings, slang, and grammatical mistakes, including the misinterpretation of ironic or sarcastic statements. Moreover, (sv)VADER ignores the context of the words it analyzes, particularly when word order and discontinuous structures involve cases where the insertion of e.g., one or more lexical items, appears between a lexicalized multiword entry and at a longer distance than the very near context.

## 4   Conclusions and future work

VADER offers a simple process for sentiment classification with a design focus on social media texts, where no training data is required, and can be used as a baseline method to evaluate and compare other methods. In this paper we outlined the scaling-up process for a dictionary approach to Swedish sentiment analysis using the VADER, a less resource-consuming lexicon and rule-based sentiment analysis tool that consumes fewer resources as compared to learning models as there is no need for vast amounts of training data. As such, VADER can serve as a good starting point to sentiment analysis before diving into more advanced machine learning (e.g., transfer learning; Prottasha et al., 2022); semiautomatic lexicon based (Chanlekha et al., 2018; Barriere & Balahur, 2020) or deep learning models[12] which stand out in terms of usage the last years, and compare their results (Dang et al., 2020). For higher level of accuracy, it may be worth evaluating alternatives (Farah & Kakisim, 2023) or even better a combination of alternative models using the VADER's sentiment scores as input feature to ensemble learning (Kazmaier & vanVuuren, 2022).

The performance of svVADER was further evaluated on two, rather small, but characteristic Swedish social media datasets. One that contains 200 tweets and one with 200 single-sentenced posts from Flashback and the achieved results were adequate. E.g., compared to GerVADER: F1-score=39,42% on German human labelled Tweets and VADER-IT, Gynaecology reviews, with F1-score=50.47%. We have also shown several ways to augment and expand the resources, and there is a strong indication in which MWEs can slightly contribute to the improvement of the results (semantic orientation) of the texts. Perhaps evaluation on much larger and varied datasets could achieve better performance.

## Limitations

There are many challenges with this approach. The representativity of the tweets or the social media sample, and their size is low and polarized, further experimentation is necessary on larger, manually curated datasets to verify the efficacy of the tool and resources on different domains and text genres. Apart from the text selection process, this paper didn't provide a comparison with learning methods, a task we left for future research.

---

[11]https://github.com/cardiffnlp/xlm-t/blob/main/src/evaluation_script.py

[12]A starting point could be the Swedish BERT models for sentiment analysis: Recorded Future & AI Sweden https://huggingface.co/RecordedFutur e/Swedish-Sentiment-Fear. The two models are based on the KB/bert-base-swedish-cased model (https://huggingface.co/KB/bert-base-swedish-cased) and have been fine-tuned to solve a multi-label sentiment analysis task.

## Acknowledgments

This work has been supported by the National Language Bank of Sweden and the HUMINFRA infrastructure project, both funded by the Swedish Research Council (2017-00626 & 2021-00176) and the project Rumour Mining (MXM19-1161:1) financed by the Bank of Sweden Tercentenary Foundation (Riksbankens Jubileumsfond).

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
