# OpenReview forum: "Scaling-up the Resources for a Freely Available Swedish VADER (svVADER)"
_NoDaLiDa/2023/Conference — NoDaLiDa 2023_

### Official Review · Reviewer_rrpA · 2023-02-27
**Adaptation to Swedish of well-known resource for sentiment analysis. Useful resource. Methodologically not terribly novel.**

**Rating:** 6
**Confidence:** 2

**Review:**

The paper describes the creation of a Swedish version of the VADER sentiment analysis dictionary and its  evaluation. In general, the paper is clear and the resource has the potential to be useful.
However, I have a number of questions:
- In what way is aspect-based sentiment analysis different from just sentiment analysis? This distinction is referred to a couple of times, notably in the description of the second dataset. However, apart from the difference in topic, the two datasets used for the evaluation do not seem to support two different kinds of sentiment analysis.
- How were the phrasal verbs that were added to the original resource collected?
- Is svVADERall in Table 2 the same as svVADER MWE in Table 1?
- The evaluation results are described as being ‘adequate’. In what sense are they adequate? Is it just because they are better than the original resource? How do they compare with evaluations of VADER for English?
- Has any error analysis been done to understand what the problems are, and why the resource performs better in one of the two datasets?

On a smaller note, the text on lines 218-219 does not parse.

**Paper Type:**

Short paper

---

### Official Review · Reviewer_5qTD · 2023-03-10
**Mainly a dataset paper with some code for rule-based classification for sentiment analysis on swedish social media data.**

**Rating:** 8
**Confidence:** 3

**Review:**

This is a data set paper, documenting an improved version of a dataset used for rule-based sentiment analysis. The main contribution is a more extensive translation and adaptation of a sentiment analysis dataset from English to Swedish. The authors claim that a large part of the corpus in English was machine translated in 2019. Though large parts, especially those without a clear translation like idioms etc, were not included in this older version. In addition to the translation and adaptation of the dataset, the main contribution, the authors also updated the codebase for the English version to include specifics for analysis of Swedish text. Inter annotator agreement is discussed and looks good. There is also no lack of examples from the translation, or discussion of difficulties.
The authors present a case study where they go into messages from twitter on mRNA vaccines and posts on immigration from a forum called Flashback (a Swedish free speech absolutist forum that all good swedes pretend is fringe, and that they would never visit). They argue that even though their method is not using state-of-the-art sentiment analysis technology, their evaluation results can be a solid and fair baseline for more modern methods. I don’t really see how adding some more modern methods would add to the paper, that work falls naturally into another paper. I’m uncertain if the dataset and code will be open source or easy to use.
The authors present some technical novelty in that they have updated and adapted the original VADER code to Swedish. Though this in itself would not merit publication, the work on translating the original corpus to Swedish is impressive. I think this publication fits the venue and I recommend a strong accept.


**Paper Type:**

Short paper

---

### Decision · Program_Chairs · 2023-03-17

Accept